# Adjuvant Transgingival Therapy with Visible Light Plus Water-Filtered Infrared-A (VIS + wIRA) in Periodontal Therapy—A Randomized, Controlled, Stratified, Double-Blinded Clinical Trial

**DOI:** 10.3390/antibiotics10030251

**Published:** 2021-03-03

**Authors:** Stefanie Anna Peikert, Anil Fischer, Anne Brigitte Kruse, Ali Al-Ahmad, Johan Peter Woelber, Kirstin Vach, Andreas Braun, Petra Ratka-Krüger

**Affiliations:** 1Department of Operative Dentistry and Periodontology, Faculty of Medicine, University of Freiburg, Hugstetter Straße 55, 79106 Freiburg, Germany; anil.fischer@googlemail.com (A.F.); anne.kruse@uniklinik-freiburg.de (A.B.K.); ali.al-ahmad@uniklinik-freiburg.de (A.A.-A.); johan.woelber@uniklinik-freiburg.de (J.P.W.); petra.ratka-krueger@uniklinik-freiburg.de (P.R.-K.); 2Practice Dr. Stefan Bertram, Gewerbegasse 5, 83395 Freilassing, Germany; 3Institute of Medical Biometry and Statistics, Faculty of Medicine and Medical Center, University of Freiburg, Stefan-Meier-Straße 26, 79104 Freiburg, Germany; kv@imbi.uni-freiburg.de; 4Clinic for Operative Dentistry, Periodontology and Preventive Dentistry, RWTH University Aachen, Pauwelsstraße 30, 52074 Aachen, Germany; anbraun@ukaachen.de

**Keywords:** antibiotic resistance, antimicrobial therapy, periodontitis, adjuvant periodontal therapy

## Abstract

The aim of this randomized, controlled, double-blinded clinical trial was to examine the additional healing effect of transgingival visible light and water-filtered infrared-A (VIS + wIRA) in the treatment of periodontitis patients compared with the standard therapy by subgingival instrumentation (SI). Therefore, forty patients with untreated periodontitis received a non-surgical periodontal treatment. Using a split-mouth study design, one quadrant of the upper jaw was randomly either exposed to VIS + wIRA four times for 20 min within two weeks in addition to SI or received only SI. Three and 6 months after intervention, clinical parameters (probing depths (PDs), clinical attachment level, bleeding on probing (BOP), furcation, tooth mobility, plaque control record, and papilla bleeding index) were re-evaluated. In the presence of PD of 4 mm and positive BOP or PD > 4 mm, SI was performed again. Moreover, the patients were asked about their discomfort using a visual analog scale from 1 to 10 for each side of the maxilla. Statistical analysis demonstrated no differences between quadrants at re-evaluation for clinical parameters (*p* > 0.05) after 3 and 6 months. Concerning pain perception, patients described less pain on the irradiated side (*p* = 0.016). In the treatment of patients with periodontitis, VIS + wIRA did not show an additional effect on the clinical outcome after 3 and 6 months. Patients described less pain on the irradiated quadrant after treatment.

## 1. Introduction

According to the Global Burden of Disease Study, the severe form of periodontitis was the sixth-most prevalent condition in the world [1]. The main etiological factor for the genesis of periodontal inflammation is a host-mediated dysbiotic bacterial biofilm. The current basic therapy of periodontitis consists mainly of the disintegration of the subgingival biofilm. This is usually achieved by removing concrements, bacteria, and toxins from the tooth surface by subgingival instrumentation (e.g., with curettes or ultrasonic instruments) [2]. Sometimes, the use of antibiotics in addition to mechanical cleaning is also indicated [3]. However, because of the increasing number of antibiotic-resistant microorganisms in the treatment of periodontitis, alternative therapy options are increasingly needed [4,5]. In other medical fields, the combination of a wide band light source with visible light wavelengths (VIS) and water-filtered infrared A (wIRA) wavelengths showed promising effects to treat various medical disorders owing to its thermal effect [6,7]. The healing effect of hyperthermia was already known to the ancient Greeks [8]. Fever is the body’s response to various stimuli of the immune system. Thus, it is also not surprising that body temperature has a strong influence on the modulation of the immune system. For example, dendritic cells stimulate and activate increased T cells at fever-like temperatures [9]. Macrophage behavior is also temperature-dependent. Thus, they showed a 40% higher phagocytosis capacity at temperatures of 40 °C than at 37 °C [10]. Besides, thermal stress leads to upregulation of toll-like receptor 4 (TLR4) expression in antigen-presenting cells (APC) and increased cytokine and NO release [11]. Increased expression of TLR4, in turn, promotes a T helper cell 1 (Th1) immune response, which has been associated with increased resistance to periodontitis [12,13,14].

Furthermore, fever-like temperatures also appear to affect microorganisms. Specifically, Gram-negative pathogens undergo enhanced lysis by blood serum after incubation at fever-like conditions [15]. *Porphyromonas gingivalis* showed an altered folding and synthesis of lipid A molecules in vitro with increasing temperature up to 41 °C, resulting in significantly enhanced activation of TLR-4 [16]. Moreover, *P. gingivalis* produced fewer virulence factors when the temperature was increased from 37 °C to 39 °C [17]. Further, fibroblasts and endothelial cells exposed to mild temperature stress (1 h at 41 °C) showed slowed cell ageing and an accelerated angiogenesis rate [18]. Direct application of heat also positively affects bone regeneration and mineralization [19,20].

VIS + wIRA is a promising possibility with a high transgingival subcutaneous tissue penetration (2–3 cm), which allows for the exposure of the periodontal pockets [7,21,22]. Besides, VIS in combination with wIRA has a low thermal surface load, allowing for the transport of high energy into tissues [23]. Furthermore, VIS + wIRA leads to an increase of the tissue temperature in the deep, oxygen partial pressure in the treated tissue and tissue perfusion, which also transports the warmth to the surrounding tissues, showing improved healing of chronic wounds after application and association with pain reduction [22,24].

All these described effects and characteristics would make VIS + wIRA interesting for adjuvant use in periodontal therapy. Therefore, this study aimed to develop a possible application form for the use of VIS + wIRA in the oral cavity, whereby a transgingival therapy of several periodontal sites can be achieved simultaneously, as well as to investigate the tissue healing effect of VIS + wIRA in the treatment of periodontal disease in the first step.

## 2. Results

A total of 46 patients were pre-examined for the study, of which 40 patients were treated with the described protocol. Six patients had to be excluded because of not appearing after successful pre-treatment/information, incorrect diagnosis in the recruitment phase, and extraction therapy by non-participating dentists. Out of 40 patients treated with VIS + wIRA, a total of 39 could be re-examined after 3 months. One patient had to be excluded owing to “repeated absence at the follow-up appointment”. After 6 months, 31 patients could be re-examined. Seven did not appear for the follow-up appointment. One patient received a subgingival instrumentation outside the study protocol in the meantime and one patient’s data had to be excluded because of a storage error (Figure 1).

No adverse events were recorded during the study period.

At baseline, 22 men (55%) and 18 women (45%) were examined, with an average age of 54.43 years. Eleven (27.5%) participants were classified as smokers. Demographic data at baseline are presented in Table 1.

At baseline, there were no significant differences between the control and the VIS + wIRA treated sextants for pocket depths (PD), bleeding on probing (BOP), clinical attachment level (CAL), plaque control record (PCR), and papilla bleeding index (PBI) (Table 2).

### 2.1. Plaque Control Record (PCR)

The average PCR related to the full dentition deteriorated with a longer period between pre-treatment from 17.47% at baseline to 19.80% after 3 months and 22.01% after 6 months. Referring to teeth 4 to 8 with PD ≥ 4 mm to ≤7 mm, a difference of −0.59% compared with baseline was seen on the VIS + wIRA exposed side after 3 months. After 6 months, a difference of 4.97% versus baseline was observed. Overall, in the inter-group comparison, no significant difference was found between the two treatment methods (after 3 months, *p* = 0.375; after 6 months *p* = 0.833).

### 2.2. Papilla Bleeding Index (PBI)

The average PBI related to the full dentition improved with a longer time interval to pre-treatment from 41.70% at baseline to 26.41% after 3 months and 19.14% after 6 months. Furthermore, the sextant treated with VIS + wIRA (teeth 4–8 with PD ≥4 mm and ≤7 mm in the upper jaw) showed a tendency toward increased reduction of PBI at the third (*p* = 0.173) and sixth (*p* = 0.265) monthly findings in the inter-group comparison, but without being statistically significant.

### 2.3. Probing Depths (PDs)

In the inter-group comparison, no significant reduction of the PDs in the VIS + wIRA treated sextant (teeth 4–8 in the maxilla with PDs between ≥4 mm and ≤7 mm) could be found after 3 (*p* = 0.453) and 6 months (*p* = 0.278) compared with the untreated side.

As the level of PD reduction also depends on the initial PD, teeth 4–8 with low PDs between ≥4 mm and ≤5 mm and teeth 4–8 with moderate PDs between ≥6 mm and ≤7 mm were evaluated separately. However, no significant difference between the two treatment methods could be observed. 

Only by evaluating the PD reduction for teeth 1–8 (ST ≥ 4 mm and ≤ 7 mm) was a significantly higher reduction determined in the quadrant treated with VIS + wIRA for the third month finding (*p* = 0.047). However, no significance was found at the sixth month finding.

### 2.4. Bleeding on Probing (BOP)

Concerning BOP, there was no significant difference between the VIS + wIRA treated sites and the unexposed sites after 3 (*p* = 0.639) and 6 months (*p* = 0.991) in the inter-group comparison.

### 2.5. Clinical Attachment Level (CAL)

The results for the third and sixth months showed a tendency toward a higher level of attachment gain in the VIS + wIRA treated sextants compared with the control (0.041 mm; 0.023 mm); however, the difference of this inter-group comparison is not significant (*p* = 0.708; *p* = 0.794).

### 2.6. Furcation

The changes in furcation were compared for treated teeth 4, 6, 7, and 8 (ST ≥ 4 mm and ≤ 7 mm) in the maxilla. The sites were considered separately (mesial, distal, and vestibular). Neither the third nor the sixth month findings showed a significant difference between the two treatment methods.

### 2.7. Tooth Mobility

No statistically significant difference between VIS + wIRA sites and the control sites could be found at the time of follow-up examinations after 3 (*p* = 0.519) and 6 months (*p* = 0.638).

All results for PCR, PBI, PD, BOP, CAL, furcation, and tooth mobility are summarized in Table 2 (third and sixth month findings). In order to compare the two methods after both 3 and after 6 months, linear mixed models adjusted for sex and age were used. However, both factors had no influence.

### 2.8. Temperature Measurement

The temperature measurement was carried out on a total of 37 test participants. Three patients had to be excluded because of technical problems with the temperature probe. The mean measured sublingual temperature was 36.13 °C. Furthermore, the temperatures in the control sextants and in the sextants treated with VIS + wIRA were measured before and after 20 min of VIS + wIRA treatment (Table 3). Temperature in the exposed sextant increased significantly (3.73 °C) compared with the control sextant (0.68 °C). Additional infrared sample images were taken before and after the VIS + wIRA treatment (Figure 2). Here, it became clear that the VIS + wIRA treatment mainly affects the treated sextant, but the remaining facial area including adjacent structures is also warmed up.

## 3. Discussion

The aim of this randomized, controlled, double-blinded clinical trial was to examine the additional healing effect of transgingival visible light and water-filtered infrared-A (VIS + wIRA) in the treatment of periodontitis patients compared with the standard therapy by subgingival instrumentation. The collected data did not show significant differences between subgingival instrumentation followed by the exposure by VIS + wIRA and subgingival instrumentation alone for the clinical outcome at 3 and 6 months after treatment. However, assessing the pain perception by visual analog scale (VAS) patients reported reduced pain on the irradiated quadrant.

The adjuvant therapy of periodontitis with VIS + wIRA has not yet been described in the literature; therefore, the results are not comparable to those of other studies, although evidence for the treatment with VIS + wIRA could be obtained from other medical fields, such as surgery or dermatology [25,26,27]. It is already known thanks to other studies that VIS + wiRA enhances the surrounding tissues and promotes their biological function. Studies have shown that fibroblasts and endothelial cells exposed to mild temperature stress (1 h at 41 °C) demonstrated slowed cell aging and an accelerated angiogenesis rate (Rattan et al., 2007). Furthermore, direct application of heat positively affects bone regeneration and mineralization and increases osteogenetic differentiation [19,20,28].

Another research group investigated the effect of increasing temperature up to 41 °C on the behavior of rat dental follicular stem cells. It was shown that, with increasing temperature up to 40 °C, these not only showed a higher cell division rate, but also increased osteogenetic differentiation, which was additionally accompanied by increased bone mineralization from 39 °C onwards. This suggests that VIS + wiRA may also have a positive effect on periodontal tissues.

However, in contrast to this study, most patients in the studies already conducted in the medical field were treated as inpatients 1–2 times a day for several weeks for 20–30 min with VIS + wIRA [7,25,27,29]. Within the scope of these above-mentioned studies, the wound healing promoting effect as well as the prevention or decrease of wound infections could be demonstrated. In contrast, our study showed a pain reducing effect of VIS + wIRA, but no apparent impact on the healing process of the periodontal inflammation.

In addition, tissue temperature increased following exposure with VIS + wIRA after 20 min to 39.01 °C, indicating that a sufficient amount of energy was transferred to the surrounding tissue in order to show possible wound healing stimulating effects. This corresponds to previous studies describing beneficial wound healing effects for a temperature range between 37 and 39 °C [22,29]. In contrast, only a slight temperature increase (+0.68 °C) was observed in the unexposed areas, which did not exceed the sublingual temperature (36.14 °C). Compared with tooth irradiation with a halogen lamp, which causes a temperature increase of 5 °C after 45–60 s [30], no adverse heating effects were shown for the irradiation using VIS + wIRA for longer time periods of up to 30 min [21]. Lastly, in a forthcoming study, the temperature measurement would have to be carried out directly in the periodontal pocket in order to make an even more accurate assessment of the temperature increase caused by VIS + wIRA.

The reduction of probing depths after 3 months (0.84 mm (control); 0.89 mm (VIS + wIRA)) and 6 months (0.93 mm (control); 1.03 mm (VIS + wIRA)) correlate with the results of other studies on non-surgical periodontitis therapy [31,32,33,34]. The strongest reduction in probing depths can be expected 1–3 months after intervention; however, further healing processes were proven up to 9–12 months after treatment [35,36,37,38]. For this reason, patients in this study were re-evaluated after 3 and 6 months, also observing a further reduction of probing depths between the third and sixth month findings. Regarding the BOP, this study showed a reduction of 30.25% and 41.63% for the third and sixth month findings, respectively, for the control and 36.05% and 48.65%, respectively, for the teeth treated with VIS + wIRA. These results are consistent with those reported in the literature [39].

Similar results could also be achieved regarding the clinical attachment gain. According to Cobb et al., the average attachment gain after non-surgical periodontal therapy is about 0.55 mm for pockets from 4 to 6 mm probing depth and 1.19 mm for pockets greater than 7 mm probing depth [39]. With an attachment gain of 0.87 mm (control) and 0.91 mm (VIS + wIRA) after 3 months and 0.93 mm (control) and 0.95 mm (VIS + wIRA), respectively, at the sixth month finding with an average pocket depth of 4.58 mm at the beginning of the examinations, an excellent result could be achieved in both the VIS + wIRA group and the control group.

Furthermore, no significant difference in the degree of furcation could be observed after either 3 or 6 months between the two treatment methods. According to Reddy et al., a stagnation of furcation can only be achieved with a low degree of furcation by non-surgical periodontal therapy. However, without surgical regenerative therapy, no healing can be expected [40].

Patient comfort was higher after subgingival instrumentation when VIS + wIRA was applied adjunctively. Of course, it should be noted that the classification of pain intensity is subjective owing to the lack of a placebo group [41]. However, the results of this study correlate with previous studies on the effect of VIS + wIRA on abdominal wound healing following elective gastrointestinal surgery [22]. In a prospective, randomized, controlled, double-blind study with 94 patients after major abdominal surgery, Hartel et al., 2007 described a significant pain reduction of 13.4 mm on a visual analog scale (VAS) of 100 mm in the VIS + wIRA group (*n* = 46) compared with control (*n* = 48), where pain remained unchanged after 230 single irradiations. In addition, the need for painkillers could be reduced by 57–70%. The reason for this could be the improved tissue perfusion, which increases the removal of metabolites such as pain mediators, lactate, and bacterial toxins, as well as the improved tissue oxygen partial pressure in the area treated with VIS + wIRA [22].

The aim of this study was to investigate the clinical outcomes of applying VIS + wIRA in addition to standard therapy by subgingival instrumentation. The healing process depends on the oral subgingival biofilm, which could be influenced by this novel treatment method. This could save the use of additional antibiotics and avoid the development of antimicrobial resistance. However, no microbiological and immunological analyses were performed in this study. Future studies could further investigate the effect of the application of VIS + wIRA of the subgingival biofilm and immunological processes. Although the main etiological factor for the genesis of periodontal inflammation is a host-mediated dysbiotic bacterial biofilm, the severity of inflammation is also manifested in clinical parameters. Moreover, clinical parameters also indirectly provide information about immunologic status. In this study, no significant differences in the clinical parameters were found between standard therapy and additional treatment with VIS + wIRA, so further microbial analysis is questionable.

Furthermore, VIS + wIRA could also be used to activate a photosensitizer in the context of antimicrobial photodynamic therapy (aPDT). Promising data are already available from in vitro studies on supra- and subgingival oral biofilm with photosensitizers activated by VIS + wIRA [21,42]. Nevertheless, the frequency of use investigated in this study was sufficient for the transgingival antimicrobial photodynamic therapy with a photosensitizer [43,44,45]. Only if the energy reaching the sulcus is high enough could VIS + wIRA be used for transgingival activation of a photosensitizer, which would significantly simplify the procedure of aPDT. However, this needs to be clarified in future studies.

No adverse events were recorded throughout the study.

Although there was no placebo group, the split-mouth design provided a way to generate coherent clinical data for two different treatment regimens. However, a few limitations have been observed in the study. The follow-up was conducted over a limited period of 6 months. Therefore, no statement can be made about long-term effects on the clinical outcome for the control and VIS + wIRA group.

## 4. Materials and Methods

The study was conducted as a split-mouth designed, randomized, controlled, stratified, double-blinded clinical trial with patients undergoing a non-surgical periodontal therapy (NST).

### 4.1. Inclusion Criteria

Age ≥ 18 years;Untreated periodontal disease (at least two teeth with a probing depth (PD) ≥4 mm and ≤8 mm in diverse quadrants (the following combinations were possible: I/II, I/III, II/IV);No systematic periodontal treatment in the last 12 months;No antibiotic intake in the last 6 months [46];Maximum 30% smokers [47].

### 4.2. Exclusion Criteria

Surgical periodontal therapy;Antibiotic intake during the study period;Pregnancy;Serious systemic conditions (for example, HIV, immunosuppression);Patients who needed antibiotic prophylaxis prior to the periodontal examination;Special anatomical tooth position such as infraposition, strong inclination, gingival overgrowth.

### 4.3. Care-Giver and Patient Recruitment

Patients who participated in the study were recruited from November 2016 until July 2017 in the Department of Operative Dentistry and Periodontology of the University of Freiburg in Germany.

The patients were initially checked using the Community Periodontal Index of Treatment Needs (CPITN) [48]. From the value 3 or 4 in one or more sextants, the indication of periodontal treatment was given. After checking the inclusion and exclusion criteria, patients were informed about the study and asked about participation. Patients received €100 as reimbursement for participating in the study.

### 4.4. Clinical Measurements and Procedures

Before starting the study, the investigator was subjected to a calibration process. For this purpose, the investigator measured probing depths (PDs), plaque control record (PCR) [49], and papilla bleeding index (PBI) [50] on various volunteers until they corresponded to at least 75% with the results of a long-term periodontal specialist [51].

As part of the systematic therapy, each patient received periodontal pre-treatment (professional tooth cleaning in combination with personalized oral self-care instructions). Afterwards, a baseline assessment was performed by the calibrated and blinded investigator (PD in mm at six sites per tooth, clinical attachment level (CAL) in mm at six sites per tooth, bleeding on probing (BOP) in %, at six sites per tooth, furcation, tooth mobility, PCR, and PBI). In one quadrant, PDs were measured twice to calculate the intra-rater variance. Subsequently, all PDs > 4 mm were treated by hand instrumentation with curettes (Gracey curettes, Hu Friedy, Chicago, IL, USA).

The allocation of the quadrant to be treated with VIS + wIRA was randomized by the responsible statistician (STATA 14 software; StataCorp LLC, College Station, TX, USA). Sealed envelopes for the quadrant treated with VIS + wIRA for each participant were prepared according to the randomization list and were only opened after the periodontal pre-treatment and the baseline assessment to identify which quadrant is supportively treated with VIS + wIRA.

A modified VIS + wIRA radiator (Hydrosun 750 FS, Hydrosun Medizintechnik GmbH, Müllheim, Germany) with a 7 mm water cuvette was used as a radiation source. According to the manufacturer, the infrared source (USHIO, Cypress, CA, USA) delivers 15,600 lumens at a color temperature of 3050 K. The cuvette serves as a light filter and absorbs especially in the infrared B and C range. In addition, the light is filtered through a dichroic color filter BTE 595 orange (Bte Bedampfungstechnik GmbH, Elsoff, Germany), which is only transparent to wavelengths ≥595 nm. Consequently, VIS + wIRA is a special electromagnetic wave spectrum in the range of 780–1400 nm. In order to apply the light with sufficient intensity in a quadrant, it was transmitted via a light guide to individually manufactured splints (Makerbot Replicator, MakerBot Industries, Brooklyn, NY, USA). According to the manufacturer, the total power density at the end pieces is 450 mW/cm². After subgingival instrumentation of all diseased teeth with Gracey curettes, only one quadrant was additionally treated with VIS + wIRA for 20 min. Within the following 2 weeks, the treatment with VIS + wIRA was repeated four times for 20 min at a time (Figure 3).

After 3 and 6 months, the follow-up examinations were carried out. In the presence of PD of 4 mm and positive BOP or PD > 4 mm, SI was performed again.

### 4.5. Temperature Measurement

Using a temperature probe with a diameter of 0.25 mm (Testo AG, Lenzkirch, Germany, Type K t/c No. 0602.0493) and a thermometer (Testo AG, Lenzkirch, Thermometer Testo 735), the sublingual temperature and the temperature at a previously determined location in the direct radiation area in the first and third sextant of the maxilla was measured before the treatment with VIS + wIRA. After 20 min of radiation, the measurement was repeated and determined in the first and third sextant of the maxilla.

### 4.6. Pain Perception

Immediately after treatment, the patients were interviewed about their discomfort according to a visual analog scale from 1 to 10 (VAS) for each side of the maxilla [52].

### 4.7. Sample Size and Statistical Considerations

Sample size calculation and analysis of the results was conducted by a statistician (KV). It was calculated that, with a sample size of 34 patients, a difference in PD of 0.5 mm with a standard deviation of 1 can be shown with a power of 80% in a split-mouth design. Owing to dropouts, a sample size of 40 was planned.

For descriptive analysis, mean and standard deviations were computed. A paired t-test was used to compare the different parameters at baseline.

Linear mixed models were fitted to compare the two methods at both 3 and 6 months, and sex and age were used for adjustment. Pain density was compared using the Wilcoxon matched-pairs sign-test.

## 5. Conclusions

The adjuvant treatment with VIS + wIRA as part of the systematic periodontal therapy did not show an additional effect on the clinical outcomes after 3 and 6 months. However, it was possible to develop an application form for the use of VIS + wIRA in the oral cavity, whereby a transgingival thermal therapy for the periodontium of several teeth was achieved simultaneously. Consequently, future studies for the transgingival photodynamic therapy with VIS + wIRA in combination with a photosensitizer can be conducted. Furthermore, patients reported less pain on the irradiated side after treatment.

## Figures and Tables

**Figure 1 antibiotics-10-00251-f001:**
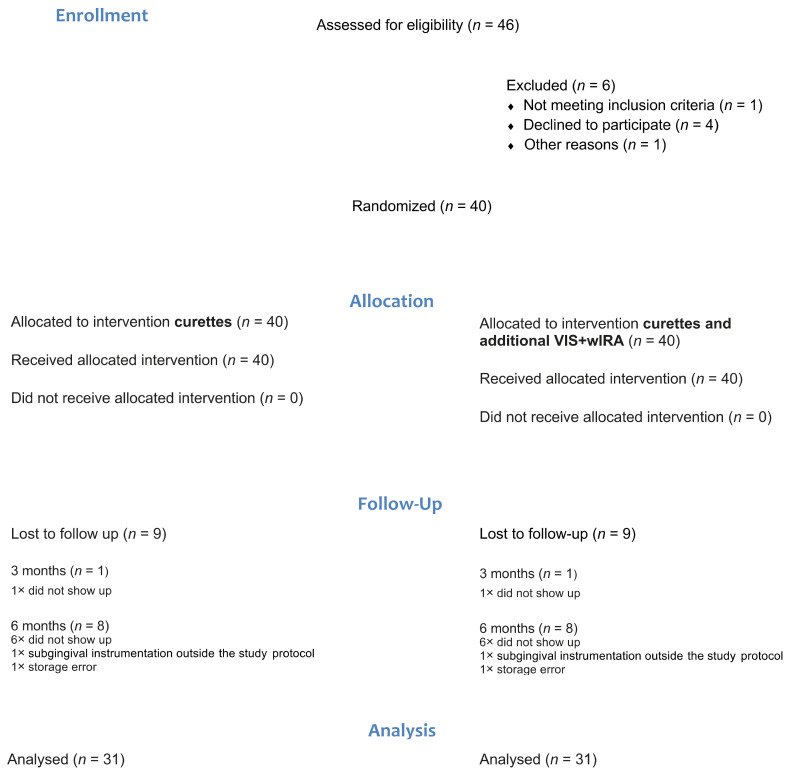
Consort diagram. VIS, visible light; wIRA, water-filtered infrared A; *n* stands for the number of study participants.

**Figure 2 antibiotics-10-00251-f002:**
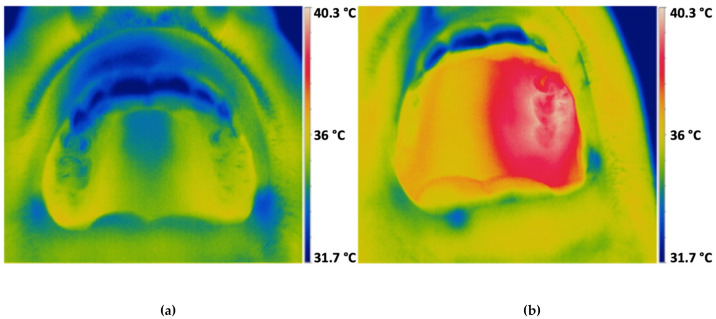
Images taken by thermographic camera before and after VIS + wIRA treatment: (**a**) infrared image taken before VIS + wIRA treatment; (**b**) infrared image taken after 20 min of VIS + wIRA treatment.

**Figure 3 antibiotics-10-00251-f003:**
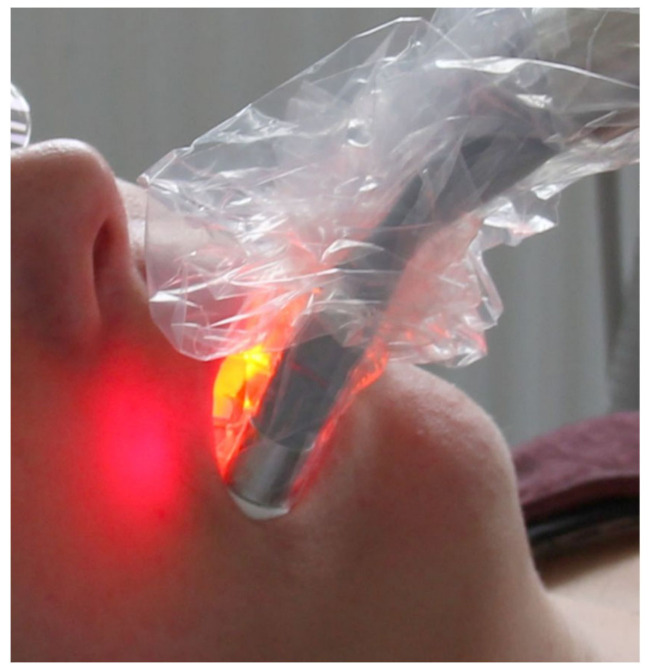
Exposure of a participant with VIS + wIRA.

**Table 1 antibiotics-10-00251-t001:** Demographic data at baseline (absolute numbers, mean (±sd), or percentages). VIS, visible light; wIRA, water-filtered infrared A.

Parameter	
Participants	40 (18 female, 22 male)
Number of treated quadrants (split-mouth)	*n* = 40 (VIS + wIRA); *n* = 40 (control)
Age, years	54.43 (± 10.57)
Smoking status	72.75% non-smokers, 27.5% smokers

**Table 2 antibiotics-10-00251-t002:** Mean values (±SD) for plaque control record (PCR), papilla bleeding index (PBI), probing depth (PD), bleeding on probing (BOP), clinical attachment level (CAL), furcation, and tooth mobility (teeth 4 to 8 with PD ≥ 4 mm to ≤7 mm) at baseline, 3 months, and 6 months for the test group (VIS + wIRA) and the control group (hand instrumentation with Gracey curettes).

Variable		Group	Baseline = BL (*n* Control = 113, *n* Test = 112)	Difference Value BL vs. 3 Months (*n* = 39)	*p*-Value	Difference Value BL vs. 6 Months (*n* = 31)	*p*-Value
PCR %		Control	21.46 (23.59)	2.92 (0.22)	*p* = 0.375	2.80 (0.21)	*p* = 0.833
		Test	22.76 (23.38)	−0.59 (0.20)		4.97 (0.26)	
PBI %		Control	39.82 (75.04)	−5.08 (0.77)	*p* = 0.173	−17.74 (0.71)	*p* = 0.265
		Test	42.85 (86.69)	−11.15 (0.79)		−23.27 (0.82)	
PD ^1^ (mm)		Control	4.58 (0.65)	−0.83 (0.62)	*p* = 0.453	−0.93 (0.53)	*p* = 0.278
		Test	4.58 (0.67)	−0.88 (0.53)		−1.03 (0.55)	
PD ^2^ (mm)		Control	4.55 (0.68)	−0,86 (0.61)	*p* = 0.047 *	−1.10 (0.86)	*p* = 0.065
		Test	4.54 (0.67)	−0,99 (0.53)		−1.22 (0.86)	
BOP %		Control	55.01 (28.68)	−16.64 (0.29)	*p* = 0.639	−22.90 (0.30)	*p* = 0.991
		Test	51.63 (28.64)	−18.61 (0.22)		−25.12 (0.23)	
CAL (mm)		Control	5.23 (1.40)	−0.86 (0.77)	*p* = 0.708	−0.92 (0.69)	*p* = 0.794
		Test	5.22 (1.33)	−0.91 (0.68)		−0.95 (0.68)	
Furcation	mesial ^#^	Control	0.23 (0.60)	−0.01 (0.33)	*p* = 0.540	0.02 (0.33)	*p* = 0.992
		Test	0.32 (0.75)	0.03 (0.22)		−0.02 (0.50)	
	distal ^#^	Control	0.59 (0.91)	−0.14 (0.60)	*p* = 0.576	−0.16 (0.66)	*p* = 0.716
		Test	0.47 (0.79)	−0.11 (0.63)		−0.13 (0.68)	
	vestibular ^#^	Control	0.56 (0.86)	−0.09 (0.56)	*p* = 0.920	−0.13 (0.59)	*p* = 0.113
		Test	0.54 (0.78)	−0.06 (0.52)		−0.17 (0.56)	
Mobility ^#^		Control	0.25 (0.58)	−0.05 (0.38)	*p* = 0.519	−0.08 (0.46)	*p* = 0.638
		Test	0.13 (0.43)	−0.02 (0.38)		−0.05 (0.40)	

* statistically significant. BL = baseline. ^1^ Calculated values for teeth 4–8; ^2^ calculated values for teeth 1–8. Data are presented as means and standard deviation, which is given in parentheses. In the baseline examination, *n* stands for the number of teeth; otherwise, *n* stands for the number of study participants. ^#^ The median is 0 in both groups at BL.

**Table 3 antibiotics-10-00251-t003:** Temperature measurement before and 20 min after VIS + wIRA treatment.

			*n*	Mean	Std. Dev.
Sublingual temperature			37	36.13 °C	0.43 °C
Temperature posterior region	0 min	Control	37	35.38 °C	0.73 °C
		VIS + wIRA	37	35.27 °C	0.74 °C
	20 min	Control	37	36.06 °C	0.58 °C
		VIS + wIRA	37	39.00 °C	0.55 °C

*n* = number of study participants, Mean = mean value, Std. Dev. = standard deviation.

## Data Availability

The datasets used and/ or analysed during this study are available from the corresponding author on reasonable request.

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
