# Peer review of "Adjuvant Transgingival Therapy with Visible Light Plus Water-Filtered Infrared-A (VIS + wIRA) in Periodontal Therapy—A Randomized, Controlled, Stratified, Double-Blinded Clinical Trial"

_antibiotics, 2021, doi:10.3390/antibiotics10030251_

Round 1

Reviewer 1 Report

The authors presented a manuscript aimed on demonstrate the additional healing effect of transgingival visible light and water-filtered infrared-A in the treatment of periodontitis patients compared to the standard therapy by subgingival instrumentation.

The subject worth investigation, however, there are important point to address before continuing the publication.

1 – The VIS + w IRA therapy is not photobiomodulation and it is definitively not antimicrobial photodynamic therapy. Thus, it is impossible to compare or make any relation between them. Remove all this comparison from the main text.

2- Another important point is that periodontal disease progression is an imbalance between overexpressed host defense against microbial colonization.  If VIS + w IRA improves host defense ability, it was not clear how it could stop or control periodontal structure loss. Please include this topic in the discussion section.

Minor points

Discussion line 177-181 – “wound healing promoting effect as well as the prevention or decrease of wound infection could be demonstrated”.  It was not clear to me if the demonstrated healing was observed in the current manuscript or in the references [7, 25, 27, 28]. Please clarify the information.

In the next sentence, the authors compare their study with antimicrobial photodynamic therapy. This comparison is difficult to proceed with since aPDT has a different mechanism of action, mainly by the formation of reactive oxygen species generated inside microbial cells.

Author Response

The authors presented a manuscript aimed on demonstrate the additional healing effect of transgingival visible light and water-filtered infrared-A in the treatment of periodontitis patients compared to the standard therapy by subgingival instrumentation.

The subject worth investigation, however, there are important point to address before continuing the publication.

  • Comment Reviewer: The VIS + wIRA therapy is not photobiomodulation and it is definitively not antimicrobial photodynamic therapy. Thus, it is impossible to compare or make any relation between them. Remove all this comparison from the main text.

Response to the reviewer: We have removed the association to photodynamic therapy out of the manuscript. (Discussion, page 6-7; Conclusion, page 10, line 390)

  • Comment Reviewer: Another important point is that periodontal disease progression is an imbalance between overexpressed host defense against microbial colonization.  If VIS + w IRA improves host defense ability, it was not clear how it could stop or control periodontal structure loss. Please include this topic in the discussion section.

Response to the reviewer: Thank you for this comment. Studies of other medical fields have shown, that VIS+wiRA enhances the surrounding tissues and promotes their biological function. We have now discussed this in more detail in the discussion section (Discussion, page 6, line 187-198).

Minor points

  • Comment Reviewer: Discussion line 177-181 – “wound healing promoting effect as well as the prevention or decrease of wound infection could be demonstrated”.  It was not clear to me if the demonstrated healing was observed in the current manuscript or in the references [7, 25, 27, 28]. Please clarify the information.

Response to the reviewer: Thank you for this remark. We have made this link more explicit in the text now. (Discussion, page 6, line 199-205)

  • Comment Reviewer: In the next sentence, the authors compare their study with antimicrobial photodynamic therapy. This comparison is difficult to proceed with since aPDT has a different mechanism of action, mainly by the formation of reactive oxygen species generated inside microbial cells.

Response to the reviewer: Thank you very much for this comment. Indeed, the link to photodynamic therapy is not appropriate in this context. We have removed this from the manuscript.    
Only if the energy reaching the sulcus is high enough, VIS+wIRA could be used for transgingival activation of a photosenitizer, which would significantly simplify the procedure of aPDT. However, this needs to be clarified in future studies. We have further discussed this aspect in the Discussion (Discussion, page 8, line 287-294).

Reviewer 2 Report

The authors present a split-mouth clinical study evaluating the clinical efficacy of the adjunctive use of transgingival therapy with visible light

plus water-filtered infrared-A (VIS + wIRA) in periodonta. The study is well desgned; Some aspects need to be addressed:

Entire manuscript: please check interpunctuation and spelling; some errors can be found.

Abstract:

  • VIS+wIRA was performed 4 tines within 2 weeks; WAS SRP performed each time?
  • Who was the control group/which quadrants and what was the treatment in the control group. Please explain this also in the abstract.
  • Please replace SRP with SI or other term. Also, please write out the abbreviation before its first use.

Material and methods:

  • Why were wisdom teeth included in the study? These are usually not included, based on their sometimes non-normal anatomical position.

Results:

  • Please correct the temperatures in the text page 161, line 156
  • Did the authors perform inter-group comparisons? This is not clear in the text and not visible in the tables

Author Response

The authors present a split-mouth clinical study evaluating the clinical efficacy of the adjunctive use of transgingival therapy with visible light plus water-filtered infrared-A (VIS + wIRA) in periodonta. The study is well desgned; Some aspects need to be addressed:

  • Comment Reviewer: Entire manuscript: please check interpunctuation and spelling; some errors can be found.

Response to the reviewer: Thank you for this comment. An English proofreading has already been done, of which a letter of confirmation was attached to the submission.

We have now rechecked the manuscript for punctuation and spelling and corrected any errors.

Abstract:

  • Comment Reviewer: VIS+wIRA was performed 4 tines within 2 weeks; WAS SRP performed each time?

Response to the reviewer: No subgingival instrumentation was performed after each treatment with VIS+wIRA. Only during the follow-up after 3 and 6 months, SI was performed after clinical parameters were reevaluated. In the presence of PPD of 4mm and positive BOP or PPD > 4mm, subgingival instrumentation was performed again. We added this information to the abstract (Abstract, page 1, line 24-25).

  • Comment Reviewer: Who was the control group/which quadrants and what was the treatment in the control group. Please explain this also in the abstract.

Response to the reviewer: Thank you for this helpful comment. We have added this information to the abstract (Abstract, page 1, line 20-22).         

  • Comment Reviewer: Please replace SRP with SI or other term. Also, please write out the abbreviation before its first use.

Response to the reviewer: Thank you for pointing this out. We have replaced the abbreviation SRP with SI. (Abstract, page 1, line 1; page 1 line 21-22; page 1, line 25)

Material and methods:

  • Comment Reviewer: Why were wisdom teeth included in the study? These are usually not included, based on their sometimes non-normal anatomical position.

Response to the reviewer: Thank you for this remark. As part of patient recruitment, we have checked the normal anatomical position of the examined teeth, otherwise they were excluded. We added this to the exclusion criteria accordingly. (Material and Methods, page 8, line 318-319) In total, only 11 wisdom teeth were included, out of a total of 1067 teeth examined, which is equivalent to a percentage of 0.01%. 

Results:

  • Comment Reviewer: Please correct the temperatures in the text page 161, line 156

Response to the reviewer: Thank you for this comment. We have revised the temperature difference of the control. (Results, page 5, line 164; Discussion, page 7, line 226)

  • Comment Reviewer: Did the authors perform inter-group comparisons? This is not clear in the text and not visible in the tables

Response to the reviewer: Thank you for this comment. We have performed only inter-group comparisons in our statistical analysis. We added this information to the text now. (Results, page 4, line 114; line 122; line 125; page 5, line 138-139; line 143)

Reviewer 3 Report

The randomized clinical trial evaluated the additional healing effect of transgingival visible light and water-filtered infrared-A (VIS + wIRA) in the treatment of periodontitis patients compared to the standard therapy by subgingival instrumentation. The results did not show significant differences between the treatments evaluated. The followings clinical parameters were evaluated: probing depths, clinical attachment level, bleeding on probing, furcation, tooth mobility, Plaque Control Record, Papilla Bleeding Index.

It is a well written article.  I have few recommendations for the authors.

Line 34, page 1 - According to the authors ”...The main etiological factor for the genesis of periodontal inflammation is a host-mediated dysbiotic bacterial biofilm...” Then, why in the study the authors did not performed immunological and microbiological analysis?

According to the Editor the topics of the “Antibiotics” is “Oral Microorganisms and Inactivation of Oral Biofilms” and “..The Special Issue will gather recent developments in emerging methods applied to inactivate biofilms in the field of oral microbiology”. https://www.mdpi.com/journal/antibiotics/special_issues/oral_biofilm .  However, the study did not envolve microbiological analysis.

- Please clarify why this study was not registered in the clinicaltrial.gov database?

- Was the study approved by the Ethics Committee? Please add the Ethics Committee Acceptance.

Author Response

The randomized clinical trial evaluated the additional healing effect of transgingival visible light and water-filtered infrared-A (VIS + wIRA) in the treatment of periodontitis patients compared to the standard therapy by subgingival instrumentation. The results did not show significant differences between the treatments evaluated. The followings clinical parameters were evaluated: probing depths, clinical attachment level, bleeding on probing, furcation, tooth mobility, Plaque Control Record, Papilla Bleeding Index.

It is a well written article.  I have few recommendations for the authors.

  • Comment Reviewer: Line 34, page 1 - According to the authors ”...The main etiological factor for the genesis of periodontal inflammation is a host-mediated dysbiotic bacterial biofilm...” Then, why in the study the authors did not performed immunological and microbiological analysis? According to the Editor the topics of the “Antibiotics” is “Oral Microorganisms and Inactivation of Oral Biofilms” and “.The Special Issue will gather recent developments in emerging methods applied to inactivate biofilms in the field of oral microbiology”. https://www.mdpi.com/journal/antibiotics/special_issues/oral_biofilm. However, the study did not envolve microbiological analysis.

Response to the reviewer: Thank your for your remark. We submitted this paper because the special issue is also presented on the homepage with the statement, that this Special Issue will also „give new diagnostic and therapeutic insights into oral diseases“. (https://www.mdpi.com/journal/antibiotics/special_issues/oral_biofilm.). Thats why we think this paper is also appropriate for this special issue.

The rationale for this study was to investigate the clinical outcomes of the use of VIS+wIRA in addition to the standard therapy by subgingival instrumentation. The healing process depends on the oral subgingival biofilm, which could be influenced by this novel treatment method without the need for additional antibiotics due to a possible enhancement of the immunological activity of the surrounding tissue. Furthermore, antibiotics should be avoided due to the development of antimicrobial resistance. However, further studies involving intensive microbial analysis are required. This point has now been discussed in the discussion section (Discussion, page 7, line 268- page 8, 286).        
Although „the main etiological factor for the genesis of periodontal inflammation is a host-mediated dysbiotic bacterial biofilm“, the severity of the inflammation is also manifested in the clinical parameters. However, in the framework of our investigations, no significant differences between the standard therapy and the additional treatment with VIS+wIRA could be detected, which is why further microbial analysis are questionable.
This is similar to the immunologic status. Here as well some clinical paramaters indirectly provide information about immunologic status. However, in the context of future studies, this issue should be investigated in more detail. In this study no immunological assey was performed (Discussion, page 7, line 268- page 8, 286).

  • Comment Reviewer: Please clarify why this study was not registered in the clinicaltrial.gov database?

Response to the reviewer: The study was registered in the German Clinical Trials Register, which is an international clinical study register (DRKS00011137). The reference to this is noted in the section Institutional Review Board Statement section (page 10, line 402- 406). 
Please see the authors' guidelines of Antibiotics: international clinical registration in the European Union is also appropriate.

  • Comment Reviewer: Was the study approved by the Ethics Committee? Please add the Ethics Committee Acceptance.

Response to the reviewer: The reference to a positive ethical vote can be found in the Institutional Review Board Statement section (page 10, line 402- 406).

Round 2

Reviewer 1 Report

The authors improved the manuscript accordingly reviewer´s suggestion.

Reviewer 3 Report

All suggested changes are incorporated and answered.